# The Lactate Receptor HCA_1_ Is Present in the Choroid Plexus, the Tela Choroidea, and the Neuroepithelial Lining of the Dorsal Part of the Third Ventricle

**DOI:** 10.3390/ijms21186457

**Published:** 2020-09-04

**Authors:** Alena Hadzic, Teresa D. Nguyen, Makoto Hosoyamada, Naoko H. Tomioka, Linda H. Bergersen, Jon Storm-Mathisen, Cecilie Morland

**Affiliations:** 1Section for Pharmacology and Pharmaceutical Biosciences, Department of Pharmacy, The Faculty of Mathematics and Natural Sciences, University of Oslo, NO-0316 Oslo, Norway; alena.hadzic@farmasi.uio.no (A.H.); aseret-1996@hotmail.com (T.D.N.); 2Department of Human Physiology and Pathology, Faculty of Pharma-Science, Teikyo University, Tokyo 173-8605, Japan; hosoyamd@pharm.teikyo-u.ac.jp (M.H.); nhtomioka@pharm.teikyo-u.ac.jp (N.H.T.); 3The Brain and Muscle Energy Group, Institute of Oral Biology, Faculty of Dentistry, University of Oslo, NO-0318 Oslo, Norway; l.h.bergersen@odont.uio.no; 4Center for Healthy Aging, Department of Neuroscience and Pharmacology, Faculty of Health Sciences, University of Copenhagen, DK-2200 Copenhagen N, Denmark; 5Amino Acid Transporter Laboratory, Division of Anatomy, Department of Molecular Medicine, Institute of Basic Medical Sciences, Faculty of Medicine, Healthy Brain Aging Centre, University of Oslo, NO-0317 Oslo, Norway; jon.storm-mathisen@medisin.uio.no

**Keywords:** HCA_1_, HCAR1, GPR81, lactate, dorsal third ventricle, choroid plexus, tela choroidea, fibroblasts, ependymal cells

## Abstract

The volume, composition, and movement of the cerebrospinal fluid (CSF) are important for brain physiology, pathology, and diagnostics. Nevertheless, few studies have focused on the main structure that produces CSF, the choroid plexus (CP). Due to the presence of monocarboxylate transporters (MCTs) in the CP, changes in blood and brain lactate levels are reflected in the CSF. A lactate receptor, the hydroxycarboxylic acid receptor 1 (HCA_1_), is present in the brain, but whether it is located in the CP or in other periventricular structures has not been studied. Here, we investigated the distribution of HCA_1_ in the cerebral ventricular system using monomeric red fluorescent protein (mRFP)-HCA_1_ reporter mice. The reporter signal was only detected in the dorsal part of the third ventricle, where strong mRFP-HCA_1_ labeling was present in cells of the CP, the tela choroidea, and the neuroepithelial ventricular lining. Co-labeling experiments identified these cells as fibroblasts (in the CP, the tela choroidea, and the ventricle lining) and ependymal cells (in the tela choroidea and the ventricle lining). Our data suggest that the HCA_1_-containing fibroblasts and ependymal cells have the ability to respond to alterations in CSF lactate in body–brain signaling, but also as a sign of neuropathology (e.g., stroke and Alzheimer’s disease biomarker).

## 1. Introduction

The cerebrospinal fluid (CSF) is a clear, colorless fluid circulating through the cerebral ventricles and perivascular space. It serves a number of functions within the central nervous system (CNS), including hydrodynamic and metabolic aspects [1]. The CSF volume circulating at any given time is about 150 mL [2]. Reduced CSF volume can impair normal brain development and function, whereas increased CSF volume can cause hydrocephalus [3]. Furthermore, the CSF is involved in distribution of nutrients, hormones, and growth factors to the brain parenchyma. These solutes are fundamental for normal brain function [2]. The CSF is also involved in the removal of waste products implicated in neurodegeneration and brain injury, for instance, as the vehicle of the brain’s “glymphatic system” [4,5]. Therefore, the volume, composition, and movement of the CSF are of the utmost physiological importance [6]. Consequently, CSF samples are used to analyze for biomarkers in neurological diseases [7]. Despite the significant role of CSF volume, composition, and movement in physiology, pathology, and diagnostics, the choroid plexus (CP)—the main structure that produces CSF—is among the least studied structures of the brain [8].

The CP is a highly vascularized tissue that is located in the cerebral ventricles [3]. In fact, this tiny brain structure—comprising 0.25% of brain volume [9]—plays a large role in nourishing and protecting the brain [10]. The vascular endothelium of the CP is highly fenestrated and lacks tight junctions between the endothelial cells, allowing for blood plasma to filter through. To gain access to the CSF, the filtered blood plasma needs to pass through a layer of epithelial cells that are connected by tight junctions. These cells, at the border to the cerebral ventricles, form a highly selective barrier between blood plasma/interstitial fluid (ISF) and the CSF, namely the blood–CSF barrier (BCSFB) [11]. The production rate of the CSF is largely controlled by the blood flow through the CP, while the exact composition of the CSF is regulated at the level of the epithelial cells. Optimal functioning of the CP-CSF system is crucial for the maintenance and function of the CNS [3].

The CP is located in the cerebral ventricles, along the rim (“limbus”) of the hemisphere where the cerebral cortex abuts on the diencephalon, and similarly where the cerebellar cortex abuts on the rhombencephalon. Here, the neural tube does not develop neuroblasts and remains as the originally single layer of cuboid neuroectodermal cells (final ependyma), resting on mesenchymal connective tissue, which carries blood vessels (final pia mater). Proliferation of the vascular mesenchyme invaginates the ependyma lined ventricle lumen, forming the CP [12]. The cerebral ventricular system is made up by four cavities—the (bi)lateral, the third, and the fourth ventricles [13]. The third ventricle is located deep in the central part of the brain, making it one of the most inaccessible regions. It is a narrow and funnel-shaped cavity situated at the midline. The third ventricle is surrounded by many vascular structures (e.g., the circle of Willis and its branches) and crossed by the massa intermedia [14,15]. The third ventricle has a roof, a floor, and four walls. The roof, which makes a gentle upward arch, consist of four layers: one layer of neural tissue (formed by the fornix), two layers of tela choroidea (formed by meningeal pia mater and ependymal cells) interconnected by trabeculae, and one layer of cerebral vasculature in the velum interpositum (the space between the two layers of tela choroidea) [14,15]. The tela choroidea of the third ventricle gives further rise to the CP, which extends into the lateral ventricles [14,15].

The CNS is sealed from the blood by two distinct barriers: the endothelial blood–brain barrier (BBB) and the epithelial BCSFB [16,17]. The neuroepithelial lining of the cerebral ventricles also provides a barrier between the CSF and the brain, but knowledge of the nature and importance of this barrier also remains sparse. These physical barriers provide selective permeability for endogenous and exogenous substances at the interfaces, hence protecting the brain parenchyma against toxicants [18]. For nutrients, such as glucose and lactate, specific transporters are present in the barriers, allowing these substances to pass through [19]. For lactate, the transport occurs through monocarboxylate transporters (MCTs), which are also known as the solute carrier family 16 (SLC16) (www.genenames.org), as facilitated diffusion, meaning that the direction of transport is dependent on the lactate gradient [20]. Lactate is produced through glycolysis and builds up in the tissue when glycolytic activity exceeds mitochondrial respiration [20]. Therefore, chronically elevated lactate in the brain is considered a pathological sign and is found in neurological diseases such as stroke [21] and Alzheimer’s disease (AD) [22]. MCTs in the BBB allow lactate flux from brain to blood [20]. The CP also expresses MCTs: MCT1 (SLC16A1) in the apical membrane of ependymal cells lining the ventricles and MCT3 (SLC16A8) in the basolateral membrane of epithelial cells [23], allowing lactate to flux between blood and CSF with its gradient. Therefore, changes in brain and blood lactate are reflected in the CSF. In response to stroke, the lactate concentration in the CSF increases from below 1.5 mmol/L to above 2 mmol/L [21]. AD is associated with higher CSF lactate levels, which are presumably caused by compromised mitochondrial function [22]. In fact, lactate levels in the CSF has been suggested as a biomarker for mitochondrial defects in the CNS [24]. Somewhat surprisingly, lactate was found to be higher in patients with mild AD than in patients with moderate-to-severe AD, although both groups showed lactate levels above what was detected in health age-matched controls [25]. In addition to being used as a metabolite, lactate can also affect brain function by activation of the hydroxycarboxylic acid receptor 1 (HCA_1_, also abbreviated as HCAR1 (www.genenames.org)) [26,27,28,29,30]. The lactate receptor HCA_1_ is an inhibitory G-protein-coupled receptor (GPCR), and activation of the receptor leads to the inhibition of adenylyl cyclase. The receptor may modulate neuronal activity through both the Gα and Gβγ subunits [31]. Previously, we showed that HCA_1_ is located on fibroblast in the pia mater, progressing along some of the large blood vessels that penetrate from the pia into the brain parenchyma [26]. Upon activation, HCA_1_ induces angiogenesis in the hippocampus and the cerebral cortex [26]. In a very recent study, we also demonstrated that activation of HCA_1_ induced neurogenesis in the subventricular zone (SVZ), but not in the subgranular zone (SGZ) [32]. Whether lactate in the CSF has access to HCA_1_, and hence may affect brain function, is not known.

In the present study, we investigated the distribution of the lactate receptor HCA_1_ in the cerebral ventricular system, including in the CP, using monomeric red fluorescent protein (mRFP)-HCA_1_ reporter mice and immunohistochemical analysis.

## 2. Results

### 2.1. Distribution of the Lactate Receptor HCA_1_ in the Cerebral Ventricular System

To investigate the distribution of the lactate receptor HCA_1_ in the cerebral ventricular system, we performed immunohistochemistry on free floating coronal sections from the mRFP-HCA_1_ reporter mouse brain by using signal-enhancing mRFP antibody. Within the ventricular system, reporter signal immunoreactivity was only detected in the dorsal part of the third ventricle, with no detectable labeling in the ventral part of the third ventricle, the fourth ventricle, or the lateral ventricles. More specifically, strong mRFP-HCA_1_ labeling was present in the vascularized network of the CP in the roof of the dorsal part of the third ventricle (Figure 1a), the tela choroidea in the roof of the dorsal part of the third ventricle (Figure 1b), and the neuroepithelial lining of the dorsal part of the third ventricle (Figure 1c,d). The mRFP-HCA_1_ labeling appeared in a cell-like pattern along the positive structures, each with a clear core, which was consistent with a lack of labeling in the nucleus. The reporter signal was not detected in these structures in the mRFP-HCA_1_ negative control mouse (data not shown).

### 2.2. The Lactate Receptor HCA_1_ in the Choroid Plexus in the Roof of the Dorsal Part of the Third Ventricle

Collagen IV is an extracellular membrane matrix (ECM) protein in the vascular basement membranes that separate epithelial and endothelial cells from the underlying tissues [33]. An antibody against collagen IV was used to outline the vasculature of the CP to explore if the mRFP-HCA_1_ positive cells were localized in, or in close proximity to, the endothelial cells of brain capillaries. As expected, the basal lamina marker collagen IV and mRFP-HCA_1_ did not co-localize. Furthermore, we did not detect a pattern where mRFP-HCA_1_ cells were localized along the luminal side of the collagen IV-positive basal lamina, as could be expected if HCA_1_ was expressed in endothelial cells. Hence, the mRFP-HCA_1_ was probably not localized within the endothelial cells in the CP capillaries (Figure 2a–d).

Vimentin is an intermediate filament (IF) protein expressed in fibroblasts [34]. Immunolabeling of mature fibroblast by an antibody against vimentin produced a strong labeling of cells in the CP. mRFP-HCA_1_ was localized in some, but not all, vimentin-positive mature fibroblast in the CP (Figure 2e–h). Interestingly, there seemed to be distinct regions that contained a high number of HCA_1_-containing fibroblasts, while other regions contained fibroblast with no—or very low—levels of mRFP-HCA_1_. In a previous study [26], mRFP-HCA_1_ was described in platelet-derived growth factor receptor beta (PDGFR-β) positive cells in the hippocampus, which were suggested to be immature pericytes based on their location and morphology. In the present study, mRFP-HCA_1_ was also found to be localized in cells that were positive for PDGFR-β. PDGFR-β is expressed by developing smooth muscle cells, similar to pericytes, but also by myofibroblasts, which are smooth muscle-containing immature fibroblasts [35]. The PDGFR-β labeling, binding to a plasma membrane-bound protein, was not expressed in the cytosol, but it rather outlined the mRFP-HCA_1_ positive cells and some (mRFP-HCA_1_-negative) smaller fragments/processes (Figure 2i–l).

Pericytes are known to have an important role in controlling cerebral blood flow [36,37], and in the CP, this would regulate the rate of the CSF production. Given the co-localization of mRFP-HCA_1_ with PDGFR-β, which may suggest the presence of HCA_1_ in immature pericytes, a new labeling experiment was performed with mRFP-HCA_1_ and a marker for immature cells, including pericytes, such as neuron-glial antigen 2 (NG2) [38]. The NG2 antibody produced a strong cellular labeling along the outline of the CP. No co-localization with mRFP-HCA_1_ and NG2 was observed in this experiment (Figure 2m–p).

The CP contains microglia (parenchymal macrophages) that traffic into the cerebral ventricles [11]. To investigated whether mRFP-HCA_1_ was present in immune cells in the CP, immunolabeling for the microglia/macrophage marker ionized calcium binding adaptor molecule 1 (Iba1) together with mRFP-HCA_1_ was performed. Although Iba1 immunoreactivity and mRFP-HCA_1_ positive cells were localized in the same region of the CP, the two antibodies did not show any co-localization (Figure 2q–t), suggesting that immune cells of the CP do not express HCA_1_. 

Astrocytes contribute to the integrity of the CNS barriers, neurotransmitter regulation, and energy metabolism [39]. Moreover, lactate has been suggested to act on HCA_1_ of astrocytes [40]. Therefore, co-labeling for mRFP-HCA_1_ and glial fibrillary acidic protein (GFAP) was used to investigate whether mRFP-HCA_1_ was present in mature astrocytes in the CP. However, co-localization of the GFAP-positive astrocytes with mRFP-HCA_1_ positive cells was not observed (Figure 2u–x). 

### 2.3. The Lactate Receptor HCA_1_ in the Neuroepithelial Lining of the Dorsal Part of the Third Ventricle

As described in the CP, we also detected vimentin-positive cells in the walls of the dorsal part of the third ventricle, and some of the vimentin-positive cells expressed mRFP-HCA_1_ (Figure 3a–d). We further detected mRFP-HCA_1_ in PDGFR-β positive cells in the walls of the dorsal part of the third ventricle (Figure 3e–h). The co-labeling with the ependymal cells marker urate transporter 1 (URAT1) and mRFP-HCA_1_ showed an overlapping labeling pattern with the two antibodies, suggesting co-localization in the same cells. The mRFP labeling, as well as the URAT1 labeling clearly surrounded the DAPI-stained nuclei, indicating the staining of cell somata. Based on the known localization of URAT1 in the dorsal part of the third ventricle, the co-localization with this marker suggests that mRFP-HCA_1_ was localized in ependymal cells lining the ventricle wall. Interestingly, we found mRFP to be present in some, but not all URAT1-positive ependymal cells, suggesting that subtypes of ependymal cells exist, and that only some of the ependymal cells are HCA_1_ positive (Figure 3i–l).

### 2.4. The Lactate Receptor HCA_1_ in the Tela Choroidea in the Roof of the Dorsal Part of the Third Ventricle

The mRFP-HCA_1_ labeling was particularly strong in the tela choroidea. The meningeal pia mater, which contributes to form the tela choroidea [14,15], contains a thin sheet of flattened fibroblasts. These pial fibroblasts secrete vimentin [34]. As described in the CP, we also detected vimentin-positive cells in the tela choroidea in the roof of the dorsal part of the third ventricle, and most of the vimentin-positive cells expressed mRFP-HCA_1_ (Figure 4a–d). We also detected mRFP-HCA_1_ in PDGFR-β positive cells in the tela choroidea (Figure 4e–h). Ependymal cells also contribute to form the tela choroidea [14,15]. The co-labeling with the ependymal cell marker URAT1 and mRFP-HCA_1_ showed an overlapping labeling pattern with the two antibodies, suggesting co-localization in the same cells. This was observed in the tela choroidea in the roof of the dorsal part of the third ventricle (Figure 4i–l). 

## 3. Discussion

The CSF volume, composition, and movement play a significant role in the physiology, pathology, and diagnostics of the CNS [6]. Nevertheless, the CP that produces the CSF, and the neuroepithelial lining of the cerebral ventricles that separates the brain from the CSF, remain among the least studied structures of the brain [8]. Nutrients, hormones, growth factors [2], and waste products [4,5] released in the brain or transported in the blood may be found in the CSF, because a number of transporters and carrier proteins are present at the BBB and the BCSFB. Lactate, a metabolite from glycolysis, may enter the CSF through MCTs in the BBB [20] or the CP [23], and an elevated lactate level in the CSF has been suggested as a biomarker for neurological disorders [7], including stroke [21] and AD [22]. In the present study, we demonstrate that lactate in the CSF may actually directly affect cells at the border between the CSF and the brain or the blood, respectively, since these cells show an mRFP-HCA_1_ reporter signal. We demonstrate that the lactate receptor HCA_1_ protein is distributed in the dorsal part of the third ventricle. Here, it is localized in cells of the CP, the tela choroidea, and the neuroepithelial ventricular lining. The Allen Brain Atlas (https://mouse.brain-map.org/experiment/show/77464856) shows an mRNA signal for HCA_1_ that supports these findings (although at limited resolution) but indicates a more general distribution in the CP and ventricular lining. Therefore, a wider distribution of the protein might exist, although it is not detected at the sensitivity of the present method. Being a soluble protein, and having no tag for targeting to any specific cell compartment, mRFP is distributed throughout the cytoplasm of the HCA_1_-expressing cells [41]. Therefore, the localization data presented in the present study cannot show whether HCA_1_ has a polarized localization in the fibroblasts or ependymal cells. Consequently, we cannot conclude whether HCA_1_ in these cells are facing the CSF, the brain parenchyma, or both. 

Our localization of mRFP-HCA_1_ immunohistochemistry experiments places HCA_1_ in some, but not all vimentin-positive cells, presumably pia-derived fibroblast, in these structures. Some mRFP-HCA_1_ reporter signal was also found in URAT1-positive ependymal cells, as well as in PDGFR-β-positive cells, which could be immature pericytes or immature fibroblasts. HCA_1_ was not found in any of the other cells tested in our study, including in immature (NG2-positive) pericytes. The use of antibodies to identify specific cell types is always encumbered with a degree of uncertainty, as the antibodies may be unspecific, or the antigens recognized by the antibodies may be present in more than one cell type. In our experiment, we used antibodies against well-established cell type markers and that have been shown in several studies to selectively label the antigens of interest. The only exception is the URAT1 antibody, which has been extensively tested by us [42,43]. Furthermore, the interpretations of our data are based on immunolabeling as well as the morphology and localization of the cells.

Fibroblasts are known to release growth factors, including insulin-like growth factor (IGF) and vascular endothelial growth factor (VEGF) [44]. In a previous study, we reported that hippocampal VEGFA levels increase in response to the activation of HCA_1_ in mice [26]. The localization of lactate-sensing fibroblasts along the surfaces of the third ventricle, as demonstrated in the present study, opens for the possibility that these fibroblasts release growth factors in response to increased CSF lactate via HCA_1_. Data from our group demonstrate that neurogenesis in the SVZ, but not in the SGZ, is enhanced HCA_1_-dependently by lactate [32]. The HCA_1_-dependent release of growth factors into the CSF may underlie such a differential effect between the two neurogenic niches, but this has not been demonstrated experimentally.

Furthermore, the localization of HCA_1_ reported in the present study suggests that the lactate receptor is in near proximity to lactate transporters, the MCTs: MCT1 is located in the apical membrane of ependymal cells lining the ventricles, and MCT3 is located in the basolateral membrane of epithelial cells of the CP [23]. HCA_1_ localized close to the MCTs places the lactate receptor in an ideal position to sense lactate that is being released to the CSF. Since lactate transport through the MCTs follows the lactate gradient, the increase of lactate in the CSF represents an increase in lactate in the brain or blood, depending on which cells the lactate fluxes through, and receptors placed on these cells would be hit earlier and stronger than ones placed on cells not fluxing lactate. 

The ependymal cells line the cerebral ventricular system and make up the ventricular barrier [18]. Contrary to the BCSFB, the ependymal cells are joined with adherence junctions but lack occluding junctions, enabling the diffusion of CSF from the ventricular space into the brain parenchyma [18]. Their role in the brain is to support the adjacent SVZ [45,46] and, similar to the fibroblast, to produce growth factors (e.g., fibroblast growth factor (FGF), IGF, and VEGF) [46,47]. The functional or clinical relevance of HCA_1_ in the cerebral ventricular system has not been examined, but based on the localization of HCA_1_ and the known function of the cells that express the receptor, it is not unlikely that lactate-sensing fibroblasts and lactate-sensing ependymal cells may act in synergy to release growth factors in response to CSF lactate levels. 

## 4. Materials and Methods 

### 4.1. Animals 

This study was formally approved by the by the Norwegian Animal Research Authority (FOTS ID 12521 (approval date: 16 June 2017), 8243 (approval date: 8 March 2016) and 15525 (approval date: 14 August 2018)) and the Animal Use and Care Committee of the Institute of Basic Medical Sciences, Faculty of Medicine, University of Oslo. Animals were maintained under a 12:12 light/dark cycle with free access to food and water. Animal experiments were carried out by Federation of Laboratory Animal Science Associations (FELASA)-certified personnel in strict accordance with the directive 2010/63/EU of the European Parliament and of the Council of 22 September 2010 on the protection of animals used for scientific purposes. The experiments are reported in compliance with the Animal Research: Reporting in Vivo Experiments (ARRIVE) guidelines 2.0 [48].

Monomeric red fluorescent protein (mRFP)-HCA_1_ reporter mice (*n* = 5) were used in this study. The mRFP-HCA_1_ reporter mouse line was generated by inserting a cassette consisting of the mRFP under the control of the mouse HCA_1_ promoter as described [49], and it was a kind gift from Prof. Dr. Stefan Offermanns, Max Planck Institute for Heart and Lung Research, Bad Nauheim, Germany. Since mRFP is a soluble protein, cells that express HCA_1_ display mRFP throughout cytoplasm of the cell [41]. The wild-type C57BL/6 mouse, which expresses HCA_1_ but not mRFP, was used as a negative control in the labeling experiments. 

### 4.2. Immunohistochemistry 

mRFP-HCA_1_ reporter and control mice were deeply anesthetized with zolazepam 3.3 mg/mL, tiletamine 3.3 mg/mL, xylazine 0.5 mg/mL, and fentanyl 2.6 mg/mL (0.1 mL/10 g bodyweight, intraperitoneally). After the cessation of all reflexes, the mice were transcardially perfused with a fixative consisting of 4% formaldehyde (freshly made from paraformaldehyde) in 0.1 M sodium phosphate buffer pH 7.4 (NaPi) at infusion rate 5 mL/min for 8 min. Then, brains were dissected out and immersed in 30% sucrose in Milli-Q water overnight at 4 °C for cryoprotection. The brains were sectioned coronally into 20 µm thick sections at −22 °C using a freezing microtome. Immediate processing of the brains is essential, as the mRFP signal is rapidly lost.

Free-floating brain sections were rinsed (2 × 10 min) in PBS (10 mM NaPi with 135 mM NaCl), and unspecific labeling sites were blocked by incubation in a blocking solution (1% bovine serum albumin (BSA) and 3% new born calf serum (NCS) in PBS with 0.5% Triton X-100 (PBST)) for 2 h. Then, the sections were incubated with primary antibodies diluted in blocking solution overnight. The mRFP signal was enhanced by rat anti-mRFP (5F8, ChromoTek, Munich, Germany; diluted 1:500). The following antibodies were used one by one for co-labeling with mRFP: rabbit anti-collagen IV (ab6586, Abcam, Cambrige, UK; diluted:1:500), mouse anti-vimentin (code E-5, sc-373717, Santa Cruz Biotechnology, Dallas, TX, USA; diluted 1:250), rabbit anti-PDGFR-β (Y92, ab32570, Abcam, UK; diluted 1:100), rabbit anti-NG2 (AB5320, Millipore, Burlington, MA, USA; diluted: 1:200), rabbit anti-Iba1 (019-19741, Wako, TX, USA; diluted 1:500), mouse anti-GFAP (3670S, Cell Signaling Technology, Leiden, The Netherlands; diluted 1:500) or anti-mouse URAT1 [42,43]. After the overnight incubation with the primary antibodies, the sections were rinsed (6 × 10 min) in PBS and incubated with fluorescence species-specific secondary antibodies diluted in blocking solution for 2 h (shield from light). The secondary antibodies were anti-rat CY3 (712-165-150, Jackson ImmunoResearch, Sigrov, PA, USA; diluted: 1:1000), anti-rabbit Alexa Flour 488 (A21206, Invitrogen, Carlsbad, CA, USA; diluted: 1:1000), or anti-mouse Alexa Flour 488 (A21202, Invitrogen, USA; diluted: 1:1000). Then, they were rinsed (1 × 5 min) in PBS, incubated with DAPI (4′,6-diamidino-2-phenylindole, D9542, Sigma-Aldrich, St. Louis, MO, USA; diluted 1:5000 in PBS) for 15 min, and rinsed (1 × 5 min) in PBS. Finally, the sections were mounted on glass slides (Superfrost Plus, Thermo Fisher Scientific, Walsham, MA, USA) with Prolong Gold Antifade reagent (Life Technologies, Carlsbad, CA USA) and cover slipped (Corning cover glasses, Corning, NY, USA). 

### 4.3. Image Acquisition 

Images were acquired using a confocal laser scanning microscope (Zeiss LSM880—Fast Airy Scan, Carl Zeiss Microscopy, Jena, Germany) with the corresponding software (ZEN, Carl Zeiss Microscopy, Germany). The fluorescence signals were detected by focusing three laser beams onto the specimens (laser wavelengths: 405 nm, 488 nm, and 561 nm) with a small spatial pinhole for optimal optical resolution and contrast (AiryUnits: 1,8–2,8). Z-stack images were obtained through the entire thickness of the section, creating optically sectioned images (optical image thickness: 1 µm). Overview images were taken at low magnification (20×), while more detailed images were taken at high magnification (40× and 63×, as needed). The regions of interest included the CP, tela choroidea, and the neuroepithelial lining of the dorsal part of the third ventricle. The microscopy settings were adapted for each antibody but were identical for the mRFP-HCA_1_ positive reporter mice and the mRFP-HCA_1_ negative control mouse. A total of 5 mRFP-HCA_1_ positive reporter mice and 1 mRFP-HCA_1_ negative control mouse were included for each labeling experiment.

## 5. Conclusions

The lactate receptor HCA_1_ is distributed in the dorsal part of the third ventricle. It is a plasma membrane GPCR localized in the CP, the tela choroidea, and the neuroepithelial ventricular lining. This suggests that HCA_1_ may be activated by increases in lactate, released to the CSF from the ventricular and stromal space, and also gaining access to the CSF from blood (carried through the BBB and BCSFB by MCTs). Moreover, the cells expressing HCA_1_ in the dorsal part of the third ventricle are fibroblasts and ependymal cells. Both cell types have neuromodulating and neuroprotective roles. Therefore, HCA_1_ is possibly involved in the regulation of growth factor release to the CSF, and hence, in mechanisms downstream of the growth factors, under normal and pathophysiological conditions in the young and adult brain.

## Figures and Tables

**Figure 1 ijms-21-06457-f001:**
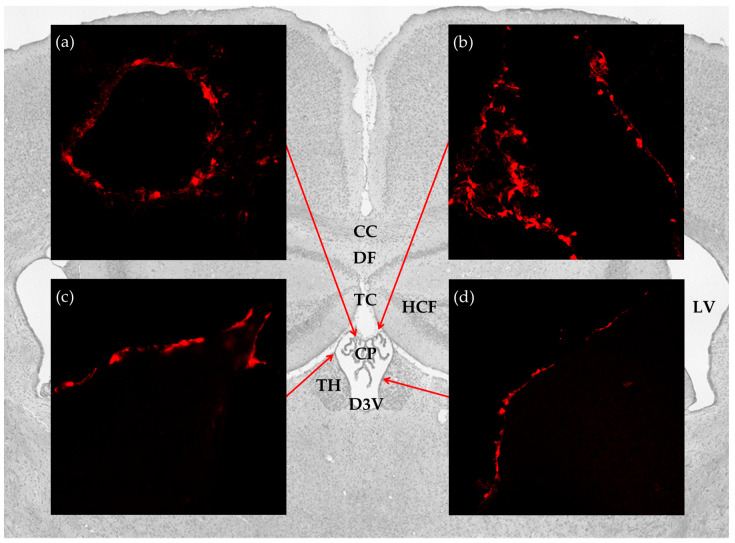
Distribution of hydroxycarboxylic acid receptor 1 (HCA_1_) in the cerebral ventricular system. Background: Image from Nissl stained coronal section of the mouse brain illustrates the anatomical localization of the brain structures studied. Insets: confocal micrographs from coronal sections of the mRFP-HCA_1_ reporter mouse brain demonstrate strong labeling in (**a**) the CP; (**b**) tela choroidea; (**c**,**d**) ventricle lining. CC: corpus callosum; CP: choroid plexus; D3V: dorsal third ventricle; DF: dorsal fornix; HCF: hippocampal formation; LV: lateral ventricles; TC: tela choroidea; TH: thalamus.

**Figure 2 ijms-21-06457-f002:**
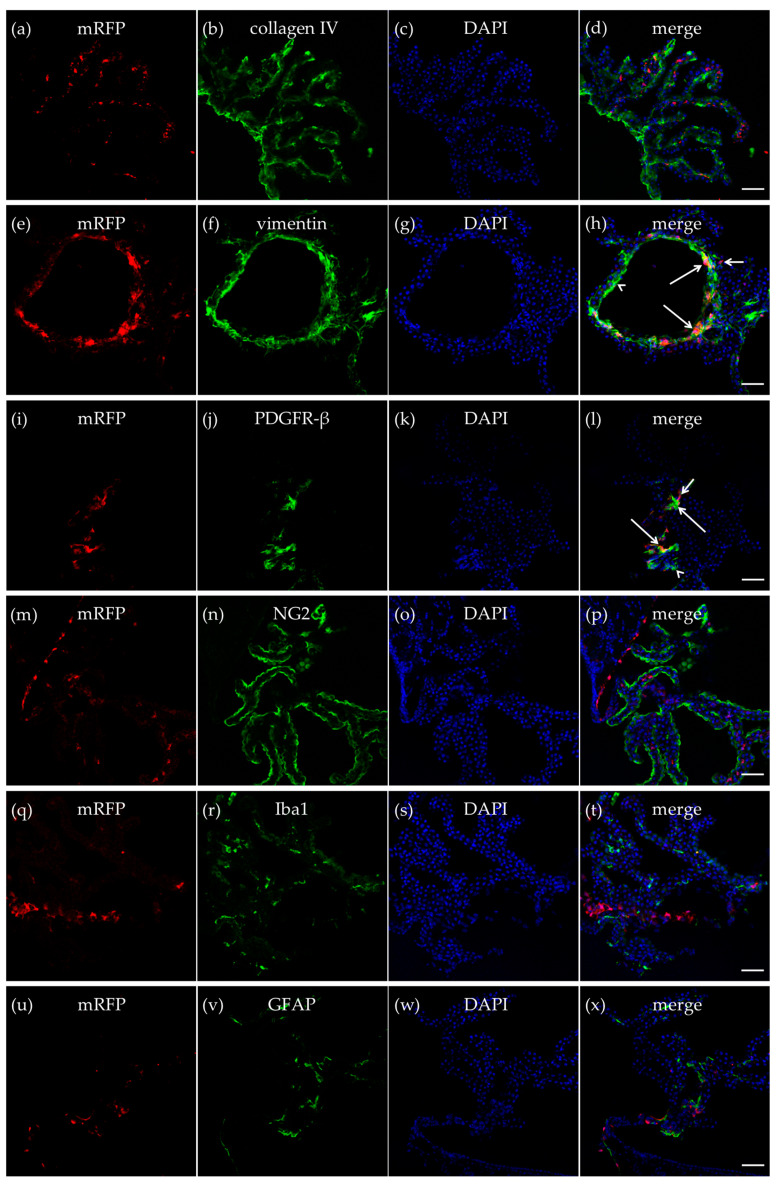
HCA_1_ in the choroid plexus. In the CP in the roof of the dorsal part of the third ventricle, mRFP-HCA_1_ (red in a-x) was found in cell-like structures surrounding a 4′,6-diamidino-2-phenylindole (DAPI)-positive nucleus (blue in **a**–**x**). mRFP-HCA_1_ did not co-localize with the basal lamina marker collagen IV (green in **a**–**d**). mRFP-HCA_1_ did co-localize with the mature fibroblast marker vimentin (green in **e**–**h**). Note that mRFP-HCA_1_ was present on some vimentin-positive cells (long arrows), but not others (arrowhead), and that some mRFP-HCA_1_ did not express vimentin (short arrow). Furthermore, some mRFP-HCA_1_ positive cells also co-localized with platelet-derived growth factor receptor beta (PDGFR-β) (green in **i**–**l**), which labels immature fibroblasts and immature pericytes. Again, we found that mRFP-HCA_1_ was present in only a subset of PDGFR-β-positive cells (long arrows) and not in others (arrowhead). Some mRFP-HCA_1_ did not express PDGFR-β (short arrow). mRFP-HCA_1_ did not co-localize with the immature cell marker neuron-glial antigen 2 (NG2) (green in **m**–**p**) which also labels immature pericytes, microglia/macrophage marker Iba1 (green in **q**–**t**), nor the mature astrocyte marker glial fibrillary acidic protein (GFAP) (green in **u**–**x**). Scale bars: 50 µm. Typical pictures shown, *n* = 5.

**Figure 3 ijms-21-06457-f003:**
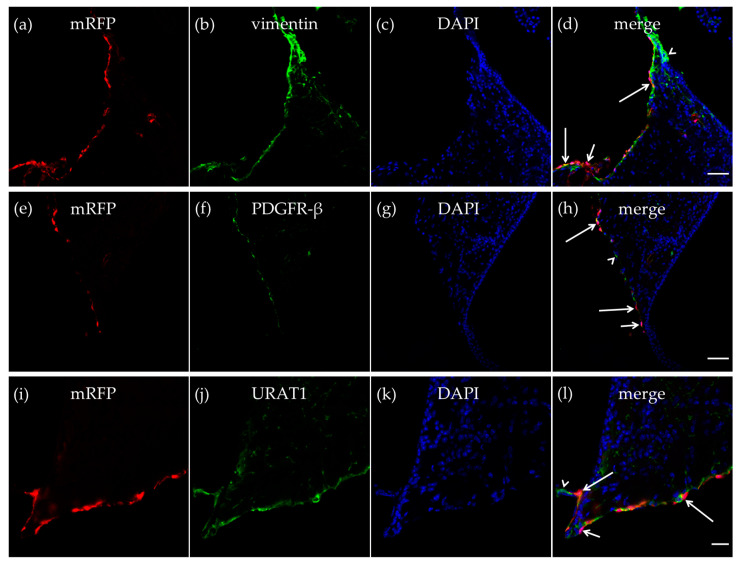
HCA_1_ in the neuroepithelial ventricular lining of the ventricle. In the neuroepithelial lining of the dorsal part of the third ventricle, mRFP-HCA_1_ (red in **a**–**l**) was found in cell-like structures surrounding a DAPI-positive nucleus (blue in **a**–**l**). mRFP-HCA_1_ positive cells co-localized with the mature fibroblast marker vimentin (green in **a**–**d**). Some vimentin-positive cells (long arrows) but not others (arrowhead) expressed mRFP-HCA_1_, while some mRFP-HCA_1_-positive cells did not express vimentin (short arrow). Furthermore, mRFP-HCA_1_ positive cells also co-localized with the immature fibroblasts and immature pericytes marker PDGFR-β (green in **e**–**h**). In addition, here, we found that mRFP-HCA_1_ was present in only a subset of PDGFR-β-positive cells (long arrows) and not in others (arrowhead). Some mRFP-HCA_1_ did not express PDGFR-β (short arrow). mRFP-HCA_1_-positive cells co-localized with the ependymal cell marker urate transporter 1 (URAT1) (green in **i**–**l**). Some URAT1-positive cells (long arrows) but not others (arrowhead) expressed mRFP-HCA_1_, while some mRFP-HCA_1_-positive cells did not express URAT1 (short arrow). Scale bars: 50 µm (**a**–**h**) and 20 µm (**i**–**l**). Typical pictures shown, *n* = 5.

**Figure 4 ijms-21-06457-f004:**
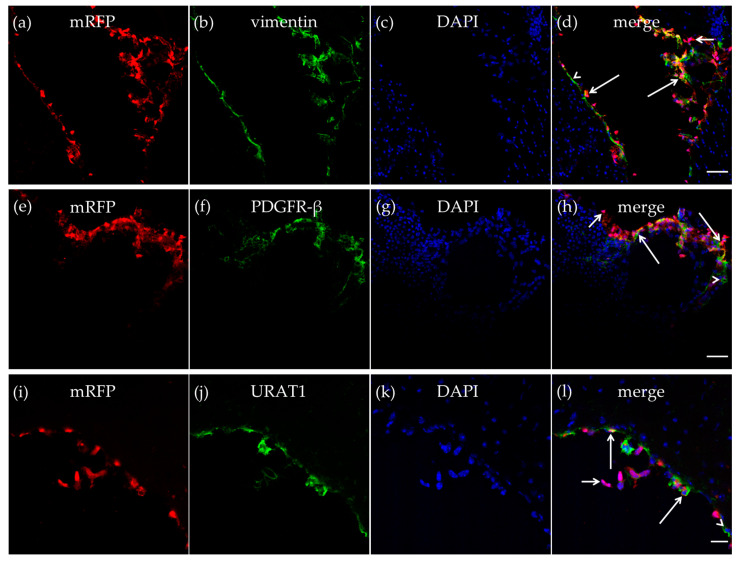
HCA_1_ in the tela choroidea. In the tela choroidea of the dorsal part of the third ventricle, mRFP-HCA_1_ (red in **a**–**l**) was found in cell-like structures surrounding a DAPI-positive nucleus (blue in **a**–**l**). mRFP-HCA_1_ positive cells co-localized with the mature fibroblast marker vimentin (green in **a**–**d**). Some vimentin-positive cells (long arrows) but not others (arrowhead) expressed mRFP-HCA_1_, while some mRFP-HCA_1_-positive cells did not express vimentin (short arrow). Furthermore, mRFP-HCA_1_ positive cells also co-localized with the immature fibroblasts and immature pericytes marker PDGFR-β (green in **e**–**h**). In addition, here, we found that mRFP-HCA_1_ was present in only a subset of PDGFR-β-positive cells (long arrows) and not in others (arrowhead). Some mRFP-HCA_1_ did not express PDGFR-β (short arrow). mRFP-HCA_1_ positive cells co-localized with the ependymal cell marker URAT1 (green in **i**–**l**). Some URAT1-positive cells (long arrows) but not others (arrowhead) expressed mRFP-HCA_1_, while some mRFP-HCA_1_-positive cells did not express URAT1 (short arrow). Scale bars: 50 µm (**a**–**h**) and 20 µm (**i**–**l**). Typical pictures shown, *n* = 5.

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
