# Peer review of "The Lactate Receptor HCA1 Is Present in the Choroid Plexus, the Tela Choroidea, and the Neuroepithelial Lining of the Dorsal Part of the Third Ventricle"

_ijms, 2020, doi:10.3390/ijms21186457_

Round 1

Reviewer 1 Report

Hadzic et al reported that HCA1-containing fibroblasts and ependymal cells have the ability to respond to alterations in CSF lactate in body-brain signaling. Data from this submission are sound. A revision is suggested.

  1. Please add the limitation of this study.
  2. Please discuss the clinical implications.

Author Response

Dear Reviewer 1,

Thank you for your kind response and constructive comments for the submitted manuscript “The lactate receptor HCA1 is present in the choroid plexus, the tela choroidea and the neuroepithelial lining of the dorsal part of the third ventricle” by Hadzic et al.

We have revised the manuscript in accordance with your comments, as specified below. In addition, we have done minor esthetical changes to our graphical abstract and improved the resolution of it.

We have corrected the affiliation for one of the co-authors (LHB), and a minor error in legend to figure 2: “arrowhead” has been changed to “arrow” (page 7, line 189).

We hope that the revised manuscript is acceptable for publication in IJMS.

Specific comments to Reviewer 1:

Hadzic et al reported that HCA1-containing fibroblasts and ependymal cells have the ability to respond to alterations in CSF lactate in body-brain signaling. Data from this submission are sound. A revision is suggested.
Response: We thank the reviewer for acknowledging the soundness of our data.

1) Please add the limitation of this study.

Response: We have now included two paragraphs about the limitation of the studies in the discussion. In the first paragraph (page 9, lines 260-264), we discuss using the fact that the mRFP reporter signal cannot be used to show the subcellular localization of HCA1. This paragraph reads: Being a soluble protein, and having no tag for targeting to any specific cell compartment, mRFP is distributed throughout the cytoplasm of the HCA1-expressing cells [41]. Therefore, the localization data presented in the present study cannot show whether HCA1 has a polarized localization in the fibroblasts or ependymal cells. Consequently, we cannot conclude whether HCA1 in these cells are facing the CSF, the brain parenchyma, or both“. Furthermore, the present study present descriptive data only, which may also be considered a limitation (see next comment).

"2 ) Please discuss the clinical implications.
Response: The present study is based entirely on descriptive data for the localization of HCA1 in the mouse third ventricle. The leap from these data, through functional investigation in animals or cells, to clinical implications is quite large. We therefore find it hard to discuss the clinical implications without risking to over-sell or stretch our data. Nevertheless, we have modified a sentence towards the end of the discussion to highlight this: “Their role in the brain is to support the adjacent SVZ [42,43] and, similar to the fibroblast, to produce growth factors (e.g. fibroblast growth factor (FGF), IGF and VEGF) [43,44]. Therefore, it is not unlikely that lactate-sensing fibroblasts and lactate-sensing ependymal cells may act in synergy to release growth factors in response to CSF lactate levels.” Has been replaced by “The functional or clinical relevance of HCA1 in the cerebral ventricular system has not been examined, but based on the localization of HCA1 and the known function of the cells that express the receptor it is not unlikely that lactate-sensing fibroblasts and lactate-sensing ependymal cells may act in synergy to release growth factors in response to CSF lactate levels.” to highlight the limitations to the data when we discuss possible functional implications. (page 10, lines 298-301)".

Furthermore, Linda Hildegard Bergersen asked to have her affiliation corrected from "Electronmicroscopy laboratory" to "The Brain and Muscle Energy group" (the rest of the affiliation is correct).

We have also added a paragraph where we discuss the usability of antibodies to detect cell-specific markers (page 9, lines 269-276). This paragraph reads: “The use of antibodies to identify specific cell types is always encumbered with a degree of uncertainty, as the antibodies may be unspecific, or the antigens recognized by the antibodies may be present in more than one cell type. In our experiment we used antibodies against well-established cell type markers, and that have been shown in several studies to selectively label the antigens of interest. The only exception is the URAT1 antibody, which has been extensively tested by us [42,43]. Furthermore, the interpretations of our data are based on immunolabeling as well as the morphology and localization of the cells”.

Sincerely (on behalf of all of the authors),

Cecilie Morland
Institute of Pharmacy
University of Oslo
Postboks 1068
Blindern 0316 OSLO

[email protected]
Office phone: +47 22844937; Cell phone +47 41547945

Reviewer 2 Report

Study investigated the presence and distribution of HCA1in CP and ventricular system in mice and identified the potential cell types carrying HCA1 in these structures.

The study is well done. The evidence presented in immages of high quality and the conclusions drawn are justified by the data.

Eventhough it is a descriptive study it provides novel knowledge concerning HCA1 based upon their distribution and location and potentially opens the door to more functional studies in the near future.

Minor:

The study would highly profit from translational data in human brain tissue. 

Author Response

Dear Reviewer 2,

Thank you for your kind response and constructive comments for the submitted manuscript “The lactate receptor HCA1 is present in the choroid plexus, the tela choroidea and the neuroepithelial lining of the dorsal part of the third ventricle” by Hadzic et al.

We have revised the manuscript in accordance with the comments made by the reviewers, as specified below. In addition, we have done minor esthetical changes to our graphical abstract and improved the resolution of it.

We have corrected the affiliation for one of the co-authors (LHB), and a minor error in legend to figure 2: “arrowhead” has been changed to “arrow” (page 7, line 189).

We hope that the revised manuscript is acceptable for publication in IJMS.

Reviewer 2

Study investigated the presence and distribution of HCA1in CP and ventricular system in mice and identified the potential cell types carrying HCA1 in these structures.

The study is well done. The evidence presented in images of high quality and the conclusions drawn are justified by the data.

Even though it is a descriptive study it provides novel knowledge concerning HCA1 based upon their distribution and location and potentially opens the door to more functional studies in the near future.

Response: We thank the reviewer for his/her positive comments to our manuscript. In particular, we are happy that the reviewer appreciates the quality of our data and the fact that they may have functional implications.

Minor:

The study would highly profit from translational data in human brain tissue. 

Response: We agree with the reviewer that it would have been interesting to investigate the localization of HCA1 in the human brain. We do not, however, have access to human material in which such studies may be performed. Furthermore, the lack of good antibodies against HCA1 complicates the translation into human brain tissue.

Sincerely (on behalf of all of the authors),

Cecilie Morland
Institute of Pharmacy
University of Oslo
Postboks 1068
Blindern 0316 OSLO

[email protected]
Office phone: +47 22844937; Cell phone +47 41547945